# Amplification of future energy demand growth due to climate change

Bas J. van Ruijven [1,2,3], Enrica De Cian[4] & Ian Sue Wing[3]

Future energy demand is likely to increase due to climate change, but the magnitude depends on many interacting sources of uncertainty. We combine econometrically estimated responses of energy use to income, hot and cold days with future projections of spatial population and national income under five socioeconomic scenarios and temperature increases around 2050 for two emission scenarios simulated by 21 Earth System Models (ESMs). Here we show that, across 210 realizations of socioeconomic and climate scenarios, vigorous (moderate) warming increases global climate-exposed energy demand before adaptation around 2050 by 25–58% (11–27%), on top of a factor 1.7–2.8 increase above present-day due to socioeconomic developments. We find broad agreement among ESMs that energy demand rises by more than 25% in the tropics and southern regions of the USA, Europe and China. Socioeconomic scenarios vary widely in the number of people in low-income countries exposed to increases in energy demand.

[1] International Institute for Applied Systems Analysis (IIASA), Schlossplatz 1, A-2361 Laxenburg, Austria. [2] National Center for Atmospheric Research (NCAR), 1850 Table Mesa Drive, Boulder, CO 80305, USA. [3] Boston University, 675 Commonwealth Ave., Boston, MA 02215, USA. [4] Ca' Foscari University of Venice and Foundation Euro-Mediterranean Center on Climate Change (CMCC), Cannaregio 873/b, 30121 Venice, Italy. Correspondence and requests for materials should be addressed to B.J.v.R. (email: vruijven@iiasa.ac.at)

Energy use is one of the human systems most directly exposed to changes in the climate[1,2]. Rising ambient temperatures are expected to increase hot season cooling demand[3] and could decrease cold season heating demand across multiple economic sectors (see definition of economic sectors for this study in Supplementary Note 1)[4,5], as well as increase agriculture's demand for irrigation during crop-growing seasons[6]. Population expansion, economic growth, shifts in the sectoral composition of economies, behavior of individuals and organizations, and the pace of technological development are multiple sources of uncertainty that will interact to determine future demand of different energy sources across regions[7]. Layered on top of this are the additional uncertainties in the timing and intensity of future temperature changes—both at the global level, driven by trajectories of global greenhouse gas emissions and radiative forcing, and at finer geographic scales, determined by the effects on future regional climates.

Identifying the relative magnitudes of these uncertainties, their shapes, and interactions is very relevant to policy makers, who need to make decisions with decadal consequences today. To characterize the interactions among these multiple sources of uncertainty, we leverage on the new socioeconomic (Shared Socioeconomic Pathway, SSPs) and climate scenarios (Representative Concentration Pathways, RCPs)[8], and combine empirically derived reduced-form responses of sectoral energy demand by fuel with climatological exposures and socioeconomic variables. The Shared Socioeconomic Pathways and Representative Concentration Pathways are a set of socioeconomic and GHG emission scenarios that have been developed by the research community at the request of the Intergovernmental Panel on Climate Change (IPCC) and that are used to make scenario-based mitigation and impacts studies more comparable across the literature.

We follow a top–down methodology as described in Swan and Ugursal (2009)[9], increasingly being used as a computationally efficient alternative[10–13] to process-based simulation models[14–17]. Engineering bottom-up models are generally applied to specific countries or regions to simulate energy performance of specific buildings or building archetypes, and to forecast specific end uses in the near term[18]. Top–down models do not articulate end-use services, but rely on aggregate national statistics and macroeconomic drivers. When extending the time horizon to the long term, bottom-up models also face the issue of technological and behavioral uncertainty, often being addressed through stylized assumptions or reduced-form relationships between the parameter of interest and income[7]. When aiming at depicting global trends, the much greater complexity and data requirements make it difficult to develop high-quality global bottom-up models[19,20], and more aggregated or statistical approaches are often preferred[18]. Top–down approaches can capture the joint influence of uncertain future climatic and socioeconomic conditions. Acknowledging parameter uncertainty and stability over time[21,22], they are increasingly being used as computationally efficient alternatives[10–13,21] to characterize global or aggregate patterns over the long run. A top–down method makes it possible to develop ex ante climate-induced potential impacts prior to any direct or indirect adjustment induced by market interactions between future technological or behavioral changes with energy supplies and the rest of the economy through price changes[23].

Our contribution is to develop climate-induced shocks that can be subsequently included in Integrated Assessment Models (IAMs) or Computable General Equilibrium (CGE) models to derive actual ex post energy consumption, which could be substantially moderated by substitution and price adjustments across multiple markets[14]. These intervening adjustments ultimately drive the economic impacts, such as changes in household incomes and welfare. Their scale and scope depend on future changes in structural, technological, and market characteristics that top–down, empirically based approaches imperfectly reflect, and that can be better analyzed using IAM or CGE frameworks[14–17].

We employ econometric estimates of the per-capita demand for three different final energy carriers associated with heating and cooling (electricity, petroleum products, and natural gas, see specific definition in Supplementary Note 1.) in four economic sectors (agriculture, industry, residential, and commercial) for tropical and temperate countries as a function of per-capita gross domestic product (GDP) and exposure to hot (>27.5 °C) and cold (<12.5 °C) days obtained from De Cian and Sue Wing (2018)[24]—hereafter DCSW. We use cold and hot days as an alternative indicator to heating and cooling degree days, commonly computed using 18 °C as the comfort threshold, a value suitable to temperate regions, but not to tropical countries. The cutoff values for defining hot and cold days in our study are based on the distribution of daily average temperature across all world regions in temperate and tropical countries and meet the need to capture low and high extremes while guaranteeing a sufficient number of observations. See Fig. 2 in DCSW[24]. We combine the income elasticities for energy consumption with GDP per-capita projections from the SSPs. We determine the climatic shocks by combining temperature elasticities with the change in the number of hot and cold days between now and 2050 from 21 Earth System Models (ESMs) and two emission scenarios: RCPs 4.5 and 8.5[25–27]. Finally, we combine the baseline projections for energy consumption with the impacts from temperature change and spatial population projections to derive the spatial patterns of changes in energy demand.

We find that vigorous (moderate) warming increases global climate-exposed energy demand before adaptation around 2050 by 25–58% (11–27%), on top of the expansion due to socioeconomic development. We find broad agreement among ESMs that energy demand increases by more than 25% in the tropics and southern regions of the United States, Europe, and China. The number of low-income people exposed to increased energy needs varies greatly across SSPs.

## Results

**Baseline energy demand around 2050.** We first combined the income elasticities for energy consumption from DCSW[24] with GDP per-capita projections from the Shared Socioeconomic Pathway[8,28,29] to construct five scenarios of baseline energy demand in 2050 without climate-change impacts, based on projections of future spatially disaggregated population[30,31] and GDP growth for 183 countries[32].

Across SSP scenarios, global population around 2050 ranges between 8.4 and 10 billion people, with modestly varying geographic distribution that concentrates the bulk of future population change in northern mid-to-low latitudes (Fig. 1a). Per-capita GDP increases from 2010's global average of $9763 to between 18,000 and $42,000, with geographic patterns of economic development differing by SSP—especially latitudinally (cf SSP4 in Fig. 1b). Combining these spatial patterns with the income elasticities (specified by energy carrier, sector, and climatic zone) results in a northward-shifted and peaked latitudinal distribution of baseline future energy demand (Fig. 1c). Across SSPs, energy demand is more sensitive to income growth as compared to population expansion, with the largest energy demand increases occurring in the rapid economic growth SSP1 and SSP5 scenarios.

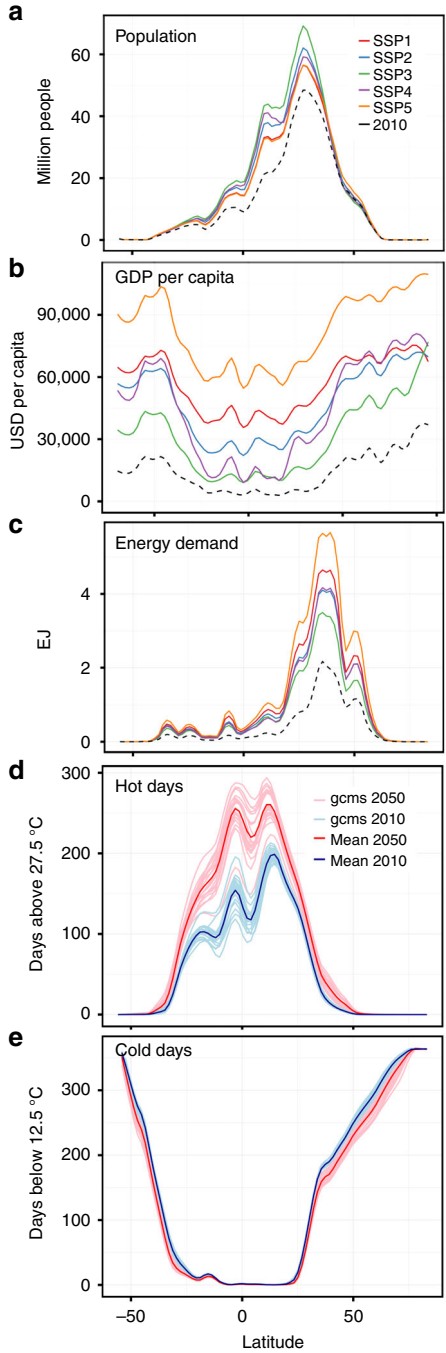

**Fig. 1** Changes in the main drivers of energy demand between 2010 and 2050. The data are smoothed and summarized by latitude: population (**a**), GDP per capita (**b**), baseline projections of the total climate-related final energy demand without climate-change impacts as a result of combining our elasticities with GDP and population projections (**c**) number of days with daily average temperature above 27.5 °C (**d**), number of days with daily average temperature below 12.5 °C (**e**) under RCP 8.5. Supplementary Fig. 6 provides a version of this figure for RCP 4.5

Current global energy demand is concentrated in the high per-capita income, high-population density areas of temperate regions, especially Western Europe, United States, Japan, and China. Baseline final energy demand grows by a factor of 1.4–2.7 for industrialized regions, with more rapid growth in China

(Table 1). Conversely, the tropics are home to developing economies that are poorer and consume far less energy, but whose population, income, and energy demand all increase significantly under the SSP scenarios. Energy demand in tropical developing regions grows by a factor of 2–4. Globally, 2050 baseline energy demand is two to three times larger than today (Table 1), or an increase from 137 EJ of climate-exposed final energy use in 2010 to 234–388 EJ by mid-century (see Supplementary Table 1).

**Impacts of climate change**. We determine the climatic shocks by combining temperature elasticities from DCSW[24] with the change between now and 2050 for spatial population and the number of of hot and cold days from the current mean climate under two emission scenarios: RCPs 4.5 and 8.5[25–27]. We use downscaled and bias-corrected simulations of all 21 Earth System Models (ESMs) that participated in the Coupled Model Intercomparison Project, Phase V (CMIP5) (see the "Methods" section)[33,34].

CMIP5 projections of temperature around 2050 show pronounced increases in hot days throughout the tropics, especially in the southern hemisphere. The median annual increase in the tropics is around 100 hot days (Fig. 1d and Supplementary Fig. 6). The mid-latitudes experience increases in the number of hot days that are much smaller, but still double their exposure compared with the present climate. Declines in exposure to <12.5 °C days are concentrated in the mid-latitudes, especially in the northern hemisphere, but these changes are much smaller in both relative and absolute magnitude (Fig. 1e and Supplementary Fig.6). Owing to the northerly skewed distribution of population, current, and future baseline energy demand, climatic changes exert impacts that are potentially large and of opposite sign compared to socioeconomic forces.

Over the period 2010–2050, the largest increases in baseline final energy demand occur in heavily populated tropical and mid-latitude areas exposed up to 75 (50) additional hot days and up to 40 (30) fewer cold days under vigorous (moderate) warming (Fig. 2 and Supplementary Fig. 7). This expansion is the largest for areas experiencing small changes in hot or cold days (<10), reflecting the fact that at mid-century, most grid cells experience only modest changes in temperature, especially under the moderate RCP 4.5 scenario. The SSP-specific multi-ESM means (Fig. 2c, d, g, h) suggest that socioeconomic forces shift this pattern monotonically, with the largest (smallest) absolute quantity of energy demand exposed under the rapid economic growth (population growth) SSP5 (SSP3) scenario. Patterns of exposure vary markedly among climate model realizations (Fig. 2a, b, e, f).

Changes in energy demand due to warmer temperatures shift the baseline exposure patterns (Fig. 2, brown lines). The total demand rises with increased frequency of hot days (with the largest effects arising for 15–70 additional days) and falls with reduced frequency of cold days (with the largest effects arising for declines of fewer than 10 cold days). The net impact is amplification of global energy demand by an amount that is an order of magnitude smaller than the socioeconomically driven increases in energy demand (Table 1), at the median of our ESM ensemble, adding 37% (20%) or 90–131 (48–67) EJ/yr by 2050 under RCP 8.5 (RCP 4.5). Across 21 ESM and 5 SSP realizations, net effects are uniformly positive but differ widely in magnitude, with a global interquartile range of 25–58% (11–27%) under RCP 8.5 (RCP 4.5).

Underlying these aggregate changes are shifts in the various sectors' demands for different energy carriers (Fig. 3 and

**Table 1 Changes in energy demand**

|  | SSP1 | SSP2 | SSP3 | SSP4 | SSP5 |
|---|---|---|---|---|---|
| *A. Ratio of 2050:2010 energy demand* | | | | | |
| Europe | 1.9 | 1.7 | 1.4 | 1.7 | 2.5 |
| North America | 1.9 | 1.8 | 1.5 | 1.8 | 2.4 |
| Oceania | 2.1 | 2.0 | 1.5 | 2.0 | 2.7 |
| South America | 2.1 | 1.9 | 1.8 | 1.8 | 2.4 |
| Middle East and Africa | 2.5 | 2.4 | 2.2 | 2.3 | 3.0 |
| Asia | 2.8 | 2.3 | 1.9 | 2.2 | 3.4 |
| World | 2.3 | 2.0 | 1.7 | 2.0 | 2.8 |
| *B. Climate-driven change in 2010–2050 energy demand growth, RCP 8.5 (%)* | | | | | |
| Europe | −1% [−5%, 4%] | 0% [−4%, 7%] | 1% [−2%, 11%] | −1% [−4%, 7%] | −3% [−6%, 1%] |
| North America | 64% [53%, 82%] | 64% [53%, 82%] | 63% [52%, 81%] | 63% [52%, 82%] | 63% [51%, 80%] |
| Oceania | 28% [19%, 41%] | 28% [19%, 41%] | 29% [19%, 41%] | 28% [19%, 41%] | 28% [18%, 41%] |
| South America | 33% [23%, 50%] | 36% [25%, 55%] | 39% [29%, 60%] | 37% [26%, 56%] | 30% [20%, 46%] |
| Middle East and Africa | 37% [29%, 57%] | 39% [30%, 58%] | 38% [30%, 56%] | 39% [30%, 57%] | 37% [28%, 55%] |
| Asia | 50% [28%, 72%] | 52% [31%, 76%] | 54% [33%, 79%] | 52% [31%, 77%] | 48% [27%, 70%] |
| World | 36% [27%, 54%] | 37% [29%, 56%] | 39% [30%, 58%] | 37% [28%, 56%] | 34% [25%, 51%] |
| *C. Climate-driven change in 2010–2050 energy demand growth, RCP 4.5 (%)* | | | | | |
| Europe | −5% [−6%, −4%] | −3% [−5%, −3%] | −3% [−4%, −2%] | −4% [−5%, −3%] | −6% [−7%, −5%] |
| North America | 31% [22%, 46%] | 31% [22%, 46%] | 31% [22%, 45%] | 31% [22%, 45%] | 30% [21%, 44%] |
| Oceania | 8% [6%, 13%] | 9% [6%, 13%] | 9% [7%, 14%] | 8% [6%, 13%] | 8% [5%, 13%] |
| South America | 15% [13%, 23%] | 16% [15%, 25%] | 19% [16%, 28%] | 17% [15%, 25%] | 13% [12%, 20%] |
| Middle East and Africa | 21% [14%, 23%] | 22% [15%, 24%] | 21% [15%, 24%] | 22% [15%, 24%] | 20% [13%, 22%] |
| Asia | 25% [15%, 36%] | 26% [17%, 38%] | 28% [18%, 39%] | 27% [17%, 39%] | 23% [14%, 34%] |
| World | 19% [12%, 25%] | 20% [13%, 27%] | 21% [14%, 28%] | 20% [13%, 27%] | 18% [11%, 24%] |

Baseline change in climate-related final energy demand by 2050 for all SSPs compared with 2010 without accounting for climate impacts (part A) and additional change in energy demand due to climate change under RCP 8.5 (part B) and RCP 4.5 (part C). Parts B and C show the median and interquartile range of all 21 CMIP5 models

Supplementary Fig. 8). The most important contributors to changes in global demand are industry and services (both net positive). The contributions of residences and especially agriculture are small and net negative. For residence, this is due to the lack of significant elasticities in our data set. For agriculture, this differs from the scarce existing literature[6], as we include multiple energy carriers that change in opposite directions. Electricity demand expands in every sector. Commercial electricity accounts for 80% of the global total climate-driven energy demand increase. Sectoral changes in natural gas demand are small and offset one another (positive in industry and negative in households). These impacts differ only slightly across socioeconomic scenarios, with households and services exhibiting the largest variation. Differences among ESM realizations (Fig. 3, error bars) are much larger. The cross-ESM distributions of demand impacts for industry, services, and the economy-wide total are long-tailed—particularly for electricity relative to other energy carriers—with a small number of climate model projections accounting for large increases in demand. The balance between increased cooling demand and decreased heating demand determines the net impact at the regional scale (see Supplementary Tables 2 and 3 for details). For Europe, the heating effect (RCP 4.5: −9 to 11%, RCP 8.5: −11 to14%) dominates the cooling effect (RCP 4.5: 4%, RCP 8.5: 10–12%), yielding a net decrease in the total final energy demand under moderate warming and a small net positive or negative impact under vigorous warming. In all other regions, climate change increases the total final energy use, with a 75th percentile impact between 45 and 75% across sectors and energy carriers in the Americas, the Middle East, and Asia under vigorous warming (Table 1).

ESMs agree that large areas of the world are likely to experience final energy demand increases of up to 10% by 2050 under RCP 8.5 and SSP5 (Fig. 4 and Supplementary Fig. 9). While increases of >50% are confined to the tropics and the southern parts of the United States, there is substantial agreement that Southern Europe and China experience increases of >25%.

There is consensus among ESMs that the total final energy use declines in Northern Europe, Russia, western Canada, and the United States, broad agreement on reductions of up to 10% in Northern Russia, the United Kingdom, and Patagonia, and no agreement that the total energy demand will decline by more than 25% in any part of the world. The differences in these spatial patterns among the SSP scenarios are minor.

**Implications for societal vulnerability and inequality.** We combine the spatial population projections for the SSPs with the changes described above to characterize the distribution of vulnerable populations to the impacts of climate change on energy demand (Fig. 5 and Supplementary Fig. 3). Impacts exhibit substantial variation across energy carriers and sectors (Supplementary Figs. 1, 2). Regarding the total final energy demand, a minority of the world's population experiences modest (0–10%) declines (mostly in Russia, Western Europe, Canada, and Chile/Argentina) and the majority faces modest (0–25%) increases, with about 30% of people experiencing substantial increases (of >25%) over a very wide range. The longest tails of the distribution are for electricity demand and for the commercial and industrial sectors. This pattern shifts only slightly across SSPs, with the biggest changes being the absolute level of the population. Higher radiative forcing lengthens the impact distribution's upper tail and increases its inter-model dispersion.

Although the aggregate population distributions of impact exposures vary only slightly across socioeconomic scenarios, they vary markedly when populations are stratified by income level (as defined by the World Bank, see Supplementary Note 2), which we take as an indicator of the capacity to adapt to changing circumstances. The exposure of populations in countries at different average per-capita income levels varies widely across the SSPs (Fig. 5; Supplementary Fig. 3). By 2050, all countries exceed the World Bank's low-income threshold; in the high economic growth SSP1 and SSP5 scenarios, almost all of the world's population lives in countries with per-capita GDP above its upper

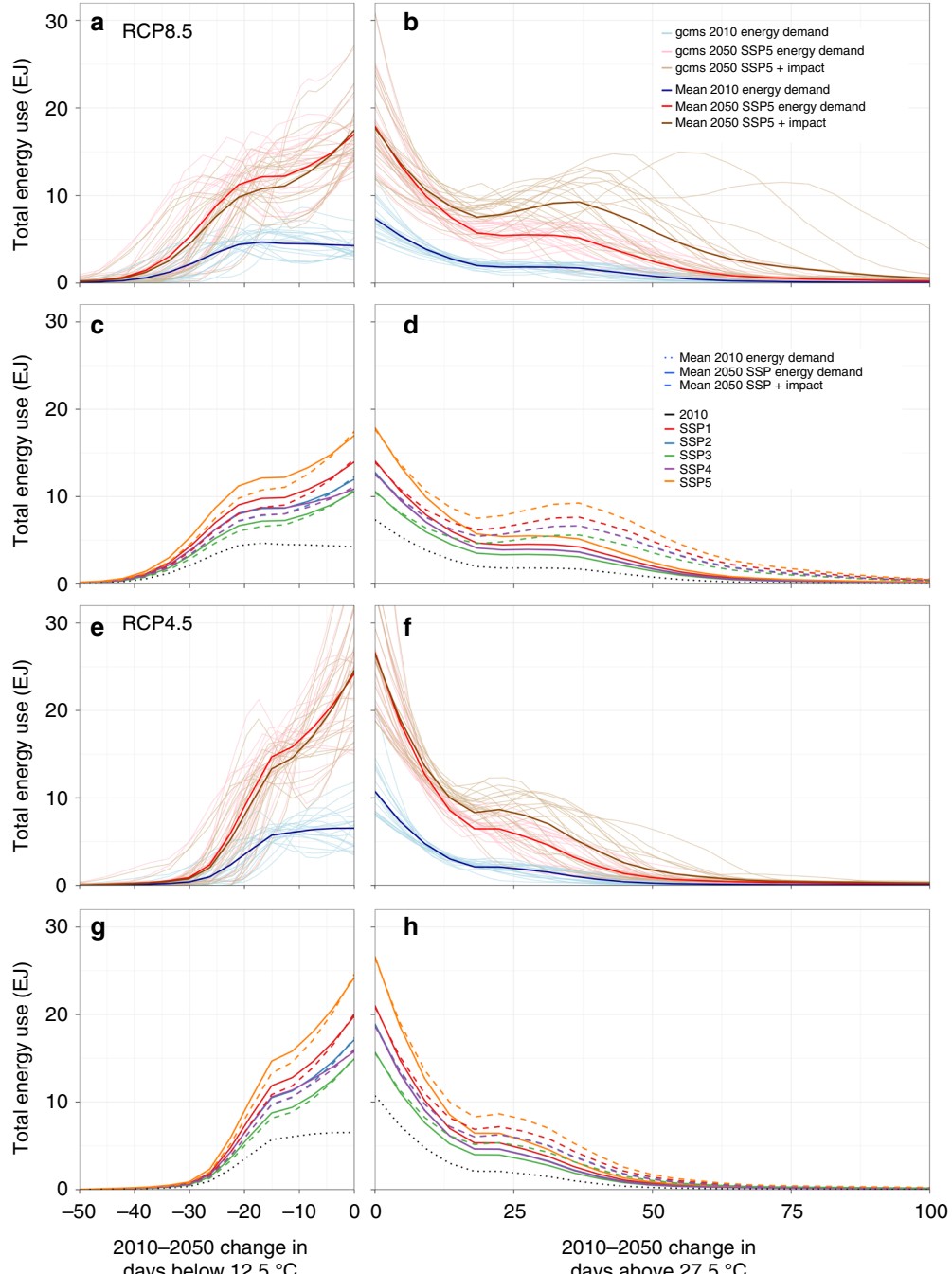

**Fig. 2** The total energy demand exposed to changes in cold and hot days. The upper half displays the results for RCP 8.5, the bottom half for RCP 4.5. In the detailed panels (**a**, **b**, **e**, **f**) the blue lines depict present-day energy demand, and the red lines depict SSP5 baseline energy demand for 2050; brown lines indicate energy use under SSP5 after impacts of climate change (mean and all 21 individual ESMs) exposed to changes in hot and cold days. Aggregate panels (**c**, **d**, **g**, **h**) show the multi-ESM mean for all five SSPs. Impacts from climate change are only shown for changes in hot (**b**, **d**, **f**, **h**) or cold (**a**, **c**, **e**, **g**) days; combined impacts are shown in Supplementary Fig. 7

middle- or high-income thresholds. Regions with a preponderance of lower middle-income countries face significant adaptation challenges under SSP2 (Asia), SSP3 (Asia, South America, Africa, and Middle East), and SSP4 (Asia, Africa, and Middle East). In the Middle East and Africa, 147–446 million inhabitants of nations facing high-adaptation challenges experience increases in the total final energy use of 25–50%, and 117–341 million experience increases of >50% under SSP2, 3, or 4. In Asia, the corresponding figures are smaller: 7–67 million and 38–66 million, respectively (Supplementary Tables 4 and 5). Moderate

warming under RCP 4.5 scales down the overall distribution of people exposed to increased energy demand.

## Discussion

Across the four economic sectors and three energy carriers, climate change increases the world's total final energy demand at mid-century by 11–27% (25–58%) under moderate (vigorous) warming on top of socioeconomic developments. As our study is the first with global comprehensive coverage across four major

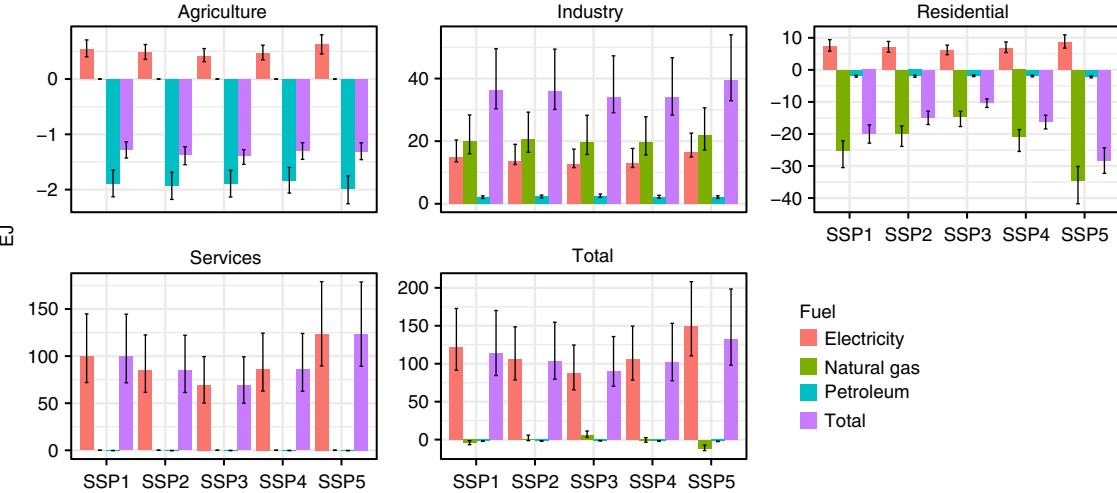

**Fig. 3** Global total energy demand amplification from climate change around 2050 under RCP 8.5. Solid bars represent the median of 21 ESM model simulations, error bars represent the interquartile range of change in energy demand across 21 ESM simulations (see Supplementary Fig. 8 for RCP 4.5 results)

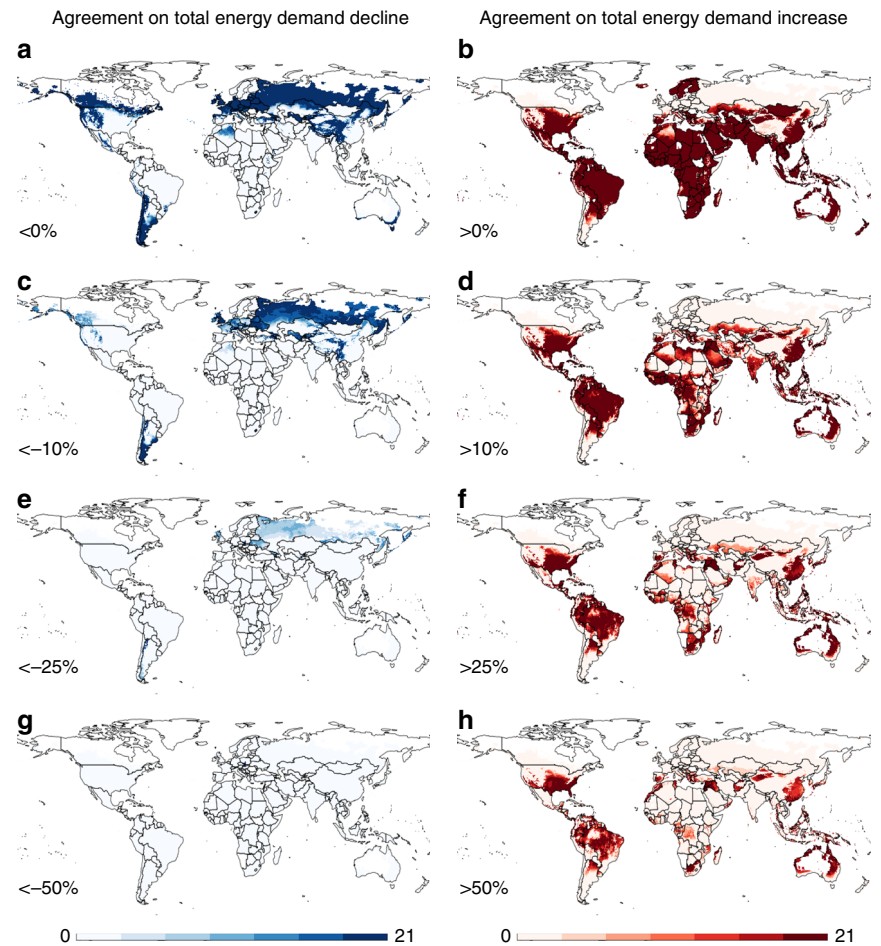

**Fig. 4** Agreement across ESMs on changes in final energy demand. Maps depict the number of climate models that agree on decreases and increases in total energy demand across sectors and energy carriers by more than 0% (**a**, **b**), 10% (**c**, **d**), 25% (**e**, **f**), or 50% (**g**, **h**) by 2050 under RCP 8.5 and SSP5, as a result of temperature projections and changes in both hot and cold days (see Supplementary Fig. 9 for RCP 4.5 results)

sectors, comparison with the literature can only be done on the sectoral or regional level. Moreover, the wide distribution across ESMs in our results implies that any comparison of future climate-change impacts based on a subset of climate models represents just a single data point in a distribution.

Starting with the sectoral dimension, the few existing global-scale studies focus on the residential sector and confirm our finding that warming will reduce overall global energy use for residential space conditioning by 2050[16,17] (though this trend reverses itself by 2100 when temperature increases are larger).

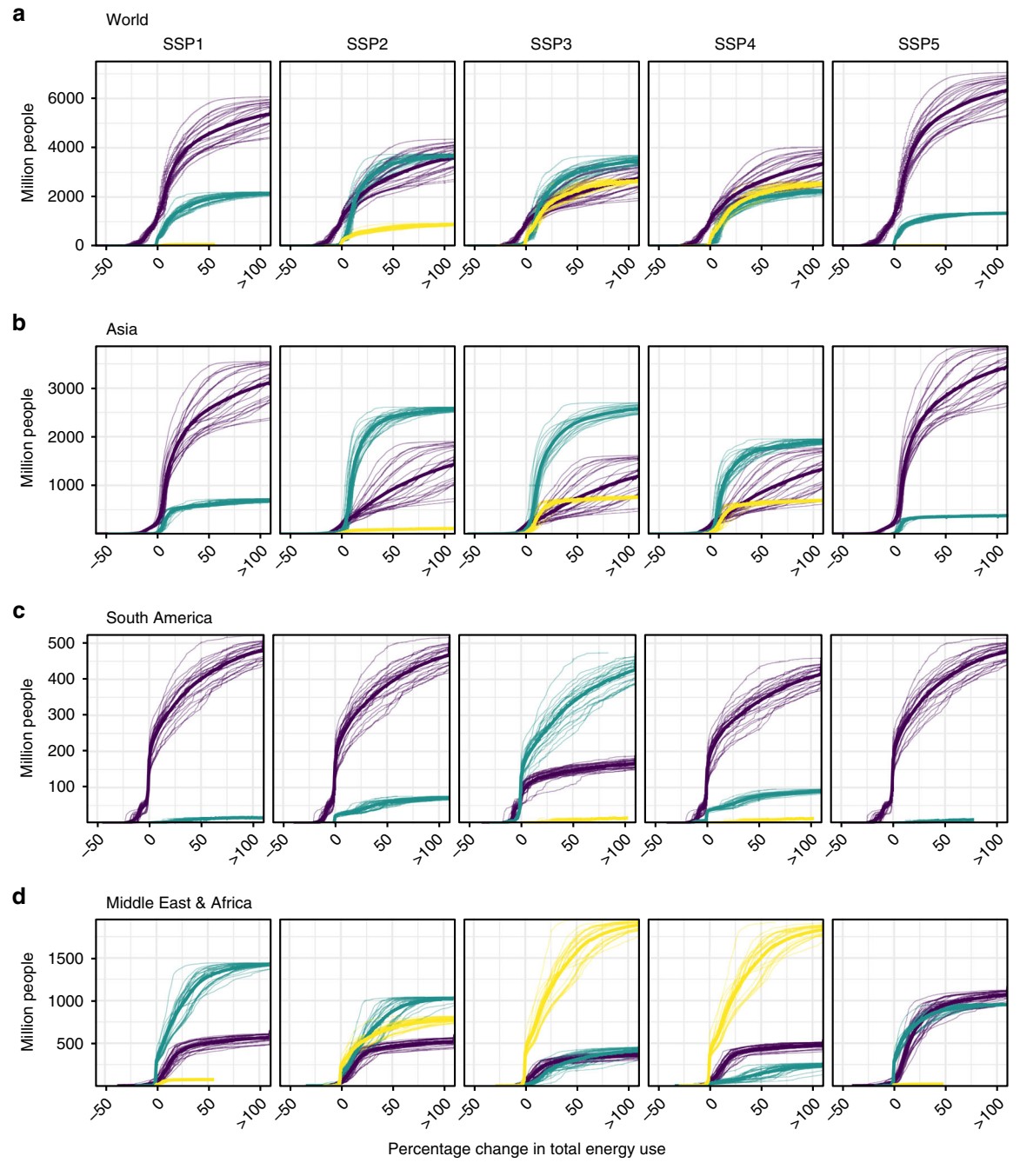

**Fig. 5** Exposure of people with different adaptation challenges to changes in energy demand. Cumulative distribution of the number of people exposed to percentage change in climate-related final energy demand by country GDP per capita. Panels represent global results (**a**) and three major developing world regions: Asia (**b**), South America (**c**), and Middle East and Africa (**d**). Lines indicate the multi-model mean and all individual 21 CMIP5 models by 2050 under RCP 8.5. Present-day World Bank definitions for GDP per capita were used to classify countries in income categories, which we used as a proxy for adaptation challenges (GDP per capita <6000 USD/yr for high-adaptation challenge, 6000–15,000 USD/yr for moderate-adaptation challenge, and >15,000 USD/yr for low-adaptation challenge, see Supplementary Note 2 for detailed discussion). Supplementary Fig. 3 presents a version of this figure for RCP 4.5 and Supplementary Figs 1, 2 present the results by sector and fuel

More recent global studies projecting future energy demand pathways for the (broader defined) building sector point at residential space cooling as a key driver for the projected net increase in final energy demand across different SSPs[7]. In our findings, the commercial and industrial sectors are the dominant drivers of energy demand increases. By considering a broader range of energy-using sectors, we highlight the potentially important impacts of commercial and industrial adaptation to climate

change, which has only been explored for specific regions[4,35–38] by few studies that confirm the potentially large impacts of these sectors.

Despite regional variation in outcomes, we find a pervasive increase in the demand for electricity to satisfy increased cooling needs in multiple sectors. For Europe, a north–south gradient of impacts of opposing sign generates a median 2% net reduction in the total final energy, on par with the findings of fuel-specific

**Table 2 Long-run estimated semi-elasticities and elasticities of energy demand to temperature and GDP per capita from DCSW[24]**

| | | Error-correction model | | | First differences | | | Static | | |
|---|---|---|---|---|---|---|---|---|---|---|
| | | Response to cold days (T < 12.5 °C) | Response to hot days (T > 27.5 °C) | Log real per-capita GDP elasticity | Response to cold days (T < 12.5 °C) | Response to hot days (T > 27.5 °C) | Log real per-capita GDP elasticity | Response to cold days (T < 12.5 °C) | Response to hot days (T > 27.5 °C) | Log real per-capita GDP elasticity |
| *Temperate regions* | | | | | | | | | | |
| Agriculture | Electricity | | 0.009 | 0.645 | | 0.001 | 0.498 | | 0.013 | −0.448 |
| | Natural gas | −0.019 | | 1.320 | 0.002 | | 1.659 | 0.002 | | 1.386 |
| | Petroleum | | | | | | | | −0.013 | |
| Commercial | Electricity | −0.006 | 0.047 | 0.864 | 0.000 | 0.001 | 0.491 | 0.001 | 0.010 | 0.458 |
| | Natural gas | | | 0.970 | | | 1.036 | | | 0.608 |
| | Petroleum | 0.012 | | −0.795 | 0.002 | | 0.825 | 0.004 | | 1.524 |
| Industrial | Electricity | | 0.009 | 0.363 | | 0.000 | 0.867 | | 0.003 | 0.539 |
| | Natural gas | | 0.033 | | | 0.000 | | | 0.014 | |
| | Petroleum | | | −1.089 | | | 1.071 | | | 0.240 |
| Residential | Electricity | | 0.015 | 0.366 | | 0.002 | 0.116 | | 0.010 | 0.376 |
| | Natural gas | 0.023 | | 1.433 | 0.002 | | 0.692 | 0.002 | | 0.963 |
| | Petroleum | 0.021 | | | 0.001 | | | 0.005 | | |
| *Tropical regions* | | | | | | | | | | |
| Agriculture | Electricity | −0.008 | | −0.701 | −0.005 | | −0.025 | −0.012 | | 0.542 |
| | Natural gas | | | | | | | | | |
| | Petroleum | 0.066 | | | 0.000 | | | 0.009 | | |
| Commercial | Electricity | | 0.008 | 0.703 | | 0.000 | 0.190 | | 0.002 | 0.415 |
| | Natural gas | | | | 0.004 | | | 0.002 | | |
| | Petroleum | −0.014 | −0.017 | | −0.001 | −0.005 | | −0.031 | −0.012 | |
| Industrial | Electricity | −0.028 | 0.008 | 0.478 | −0.002 | 0.001 | 0.496 | −0.005 | −0.002 | 0.464 |
| | Natural gas | | 0.010 | | | 0.002 | | | 0.015 | |
| | Petroleum | | 0.005 | | | 0.001 | | | 0.002 | |
| Residential | Electricity | | | 1.287 | | | 0.256 | | | 0.518 |
| | Natural gas | | | | | | | | | |
| | Petroleum | | | | | | | | | |

All estimates are significant at least at the 10% level. Insufficient observations for the missing fuel, sector, region combinations. This study uses the estimates from the error-correction model specification

studies[17,39–41]. In the United States, electricity use for cooling drives a net positive increase in final energy that, while not detected by earlier multi-fuel studies[4,37,42], is qualitatively in line with recent electricity-focused analyses[43–46]. Our disaggregated elasticities by sector and fuel also yield different results. As our elasticity for commercial electricity demand is very sensitive to hot days (Table 2), our increases in the aggregated total electricity consumption for the United States, Europe, Asia, and Latin America, exceed existing projections that use elasticities based on country-specific aggregated total electricity load[5,17,41,47,48].

Our impact projections, although comprehensive, are limited by the quantity and quality of the data available to undertake a global analysis. Similar to other existing empirical studies, we could not identify significant temperature elasticities for several combinations of energy carriers, sectors, and climatic zones (see Table 2), most importantly residential electricity use in tropical areas in response to changes in hot days. We do find that developing countries' per-capita income growth drives substantial extensive-margin investments in energy-using durable goods, particularly residential air conditioning[5,17]. However, by treating the effect of future air-conditioning penetration as orthogonal to temperature responses, our methodology ignores the possibility that this higher level of air-conditioning adoption can further amplify the growth in electricity demand with climate change[17,24,49].

An important caveat to our results is the limitations of the empirical energy demand elasticities on which we base our projections. The finest-resolution internationally comparable energy-use data are the annual records of International Energy Agency

(IEA) member countries. This presents an unavoidable trade-off between scope and fidelity, as top–down empirical climate responses constructed from these data have the geographic and sectoral coverage necessary to construct coarse-scale global projections of impacts, but struggle to replicate the fine spatial-, temporal scale of bottom-up local demand responses captured by empirical studies that employ subnational data over limited geographical domains[41,46,49,50]. In DCSW, IEA annual national fuel consumption observations in 204 countries over the years 1978–2010 are regressed on population-weighted national average exposure to numbers of days in different average temperature intervals. The resulting estimates are potentially subject to spatial and temporal aggregation bias. Large countries that span different climatic zones or incorporate regions that vary in the size and characteristics of stocks of energy-using durables (e.g., air conditioners, insulation) exhibit heterogeneous local energy demand responses to temperature. DCSW's estimates represent the mean within-country response that is a weighted average of these effects. By exploiting very large samples of high-frequency data, other studies[41,46,49,50] are also better able to capture the nonlinear response of energy demand to temperature with greater fidelity, by statistically identifying the distinct marginal effects of exposure to many temperature intervals—but over restricted geographic domains (California zip codes, US regions, Mexican municipalities, or EU countries). By contrast, IEA data's coarse resolution only permits identification of the effects of days with average temperatures of <12.5 °C and >27 °C, with each additional hot day or cold day having the same effect, irrespective of how far into the tails of the temperature it happened to be.

Notwithstanding these limitations, the demand for electricity that is a key driver of our projected impacts exhibits localized, high temporal frequency responses, whose shape can be well approximated by linear schedules outside of an intermediate moderate temperature range[46]. We exploit this property to reconcile top–down and bottom-up estimates of the response of annual electricity demand to warming (see Supplementary Note 3). Local conditional mean energy demand and the fraction of demand that is sensitive to variations in weather are crucial determinants of local impacts. However, these are not observed for the various fuel × sector combinations in grid cells across the world, and our projection methodology assumes that over the long run, all energy demand is responsive to temperature. This assumption likely overestimates the response of demand to warming, but lack of availability of local data makes it impossible to assess the magnitude and geographic distribution of the resulting bias. Our findings should thus be interpreted as worst-case projections of the impact of a given amount of warming, especially in tropical and subtropical regions that see large increases in extreme high-temperature days.

Our work shows that it is possible to identify a globally comprehensive relationship between temperature and energy, and what the potential consequences would be in the light of climatic and socioeconomic uncertainties, but also shows that better data availability would greatly enhance this work. Strengthening the empirical basis for global-scale projections of energy impact thus necessitates fundamental advances in the data collection.

Compared with impacts constructed at the mean of multiple ESM realizations of future climate change[14,41,46], our explicit consideration of uncertainty yields important insights, especially for policy makers. Although the lower tail of the RCP8.5 distribution of shocks to the global total final energy demand overlaps the upper tail of the RCP4.5 distribution (cf Table 1), Wilcoxon rank-sum tests confirm that the two densities differ. As early as mid-century, there can be benefits to mitigating climate change (t tests indicate a statistically significant 14–20% difference in the means of the distributions), but the associated costs depend critically on the uncertain greenhouse gas intensity of electricity generation that satisfies the anticipated large increases in future demand. Such increased electricity consumption does not translate one-to-one in carbon emissions, as, depending on the merit order, higher or lower carbon-intense electricity production technologies might be dominant. Similarly, adapting to an uncertain climate poses a monumental challenge to energy supply and infrastructure development planning: for RCP 8.5 (RCP 4.5), worst-case amplification among the CMIP5 ESMs of the 2010–2050 change in ex-ante total final energy demand is 150% (41%) globally, and across regions, 265% (68%), concentrated in Asia, generally in line with the recent IEA cooling demand outlook[3]. Such uncertainties, stemming from the ensemble of ESMs, dwarf the uncertainties in percentage and absolute impacts due to differences in sectoral composition of countries' energy systems under the various SSPs (Fig. 3).

Urbanization is implicit in our methodology, as future temperature fields are population-weighted, and the spatial distribution of population varies across SSPs along with country-level population trends, which do play a large role in the underlying SSP projections. Across the SSPs, urbanization levels vary widely[51], with low urbanization in SSP3 and high urbanization in SSP1 and SSP5. Urbanization drives the location of population in the SSP spatial population projections, where they are allocated from the national level to individual grid cells. In our projections, urbanization determines to which temperature changes people are exposed, and temperature changes in urban grid cells are more dominant in higher-urbanization scenarios.

Future climate change is likely to affect energy use in the transport sector as well, but as suggested in DCSW, the underlying drivers remain opaque. On the one hand, people occupying vehicles modify cooling and heating while driving to maintain thermal comfort. On the other hand, when faced with extreme cold or hot weather conditions, they might simply make different mobility choices (e.g., drive more, or do not drive at all). Weather traditionally does not enter the model of the demand for transportation, and temperature-related energy use is marginal for transportation purposes. Since our projection methodology cannot discern the fraction of energy that is sensitive to variations in weather, we focus on the final sectors where temperature more directly affects the climate-sensitive energy use. The dynamics of the transport sector are more complex and sub-sector-specific (e.g., fright transport would have different dynamics compared with personal mobility choices. In the residential sector demand for mobility, services would also interact with leisure-labor choices), calling for dedicated studies first identifying the underlying mechanisms at play, and subsequently evaluating them empirically.

We conclude by emphasizing that our results indicate the potential, ex-ante impacts of climate change, prior to any direct or indirect adaptation measures[23], representing the first step of a full-impact analysis. The present impact estimates are key inputs to the Integrated Assessment Model or CGE model simulations, which can examine the shifts in economic structure or technology between socioeconomic and climate scenarios driven by our ex ante shocks. Rebound effects and price changes will also dampen the consequent changes energy demand, further influencing the gap between our projections and ex post outcomes[52]. Given these considerations, our findings should be interpreted not as future physical changes in energy use, but as the changes in demand that, without price, substitution, and induced technical change adjustments, would maintain the level of economic services delivered by energy systems in the baseline scenario, if they were subjected to temperature shocks. Within CGE models, a practical way to operationalize such impacts could be technological shift parameters that lower the productivity of energy inputs to sectoral economic activities by the warranted percentage increase in energy demand, and vice versa. Induced technological change could then be modeled as a structural process that generates ex post modifications of such productivity shifts, following the practices of bottom-up approaches.

The emission consequences of our projected climate-driven expansion of global fossil fuel demand give rise to a fundamental tension between mitigating and adapting to higher temperatures. Follow-up investigations using IAM simulations to characterize the balance of these forces and its implications for the joint costs of mitigation and adaptation have just started to appear in the literature[53]. Empirically based, global-scale probabilistic impact projections of the kind reported here are critical to catalyzing synergistic research throughout the broader integrated assessment community.

## Methods

**Empirical approach.** Our methodology is based on DCSW[24], the relation to which is summarized in Fig. 6. DCSW[24] derived econometric estimates of per-capita demand for three different final energy carriers associated with heating and cooling in four economic sectors for temperate and tropical countries as a function of per-capita gross domestic product (GDP) and exposure to hot (>27.5 °C) and cold (<12.5 °C) days. The methodology applied in DCSW[24] is detailed in this section. This study combines their elasticities, $\beta^Y$ (income) and $\beta^T$ (temperature), with a large set of future changes in income and temperatures. In the first step, we combine the income elasticities with population and GDP projections from the Shared Socioeconomic Pathways (SSPs) to construct five scenarios of baseline global energy demand in 2050 without climate-change impacts. In the second step, we impose climatic shocks derived from 21 Earth System Models and 2 emission scenarios (RCPs) on top of the baseline scenarios by combining temperature

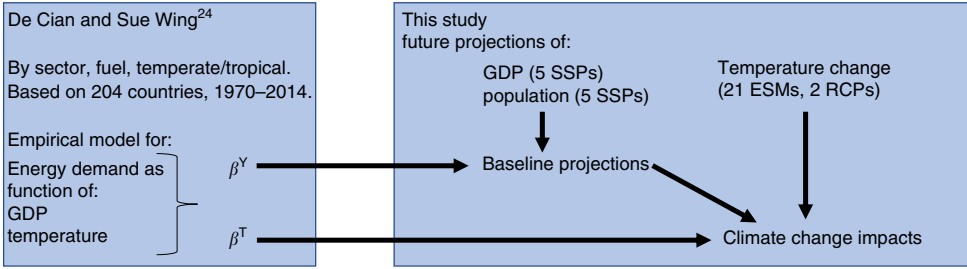

**Fig. 6** Brief overview of the methodology of this study and its relation to De Cian and Sue Wing (2018). De Cian and Sue Wing (2018) estimated elasticities of energy use with changes in temperature ($\beta_T$) and GDP per capita ($\beta_Y$). This study combines these elasticities with GDP and population projections for the SSPs to establish baseline projection and with temperature projections from 21 climate models to analyze climate change impacts

elasticities with the 2050 present difference in spatial population and hot and cold days from the current mean climate.

We use long-run income elasticities and temperature semi-elasticities estimated by DCSW[24] to explore uncertainty in projected energy demand impacts across a range of climate models and socioeconomic scenarios. DCSW[24] uses unbalanced panel data of energy demand, income per capita, prices, and weather covariates for 204 countries over the period 1970–2014 (see detailed references in DCSW[24]) to estimate income elasticities and temperature semi-elasticities of sectoral energy demand across two regions (temperate, tropical), three energy carriers (electricity, natural gas, and petroleum products), and four sectors (residential, commercial, industry, and agriculture). Differently from the customary approach used in the climate-economy literature[12], DCSW[24] models, the relationship between energy demand, weather, income, and prices as a dynamic adjustment process. The adjustment in energy demand following a shock is not immediate, because of capital fixity and adjustments in expectations. At each point in time, changes in energy demand are a function of weather, income, and price shocks, as well as of the adjustment process induced by shocks that took place in the previous period (year). Adjustments in energy use to price, income, and weather shocks occur over time, in line with studies finding evidence of persistency[12,54]. DCSW estimates the model as a dynamic panel using an Error-Correction Model (ECM), which yields short- and long-run elasticities. Over the long run, adjustments on the extensive margin (e.g., purchase of air conditioners, improvements in energy efficiency) also affect the use of energy, and this additional effect is being captured by the long-run elasticities, which we use here in this paper. The ECM can also be represented as an Autoregressive-Distributed-Lag model, and it also allows obtaining statistical inference that is robust to nonstationary data. Long-run estimates can be seen as a weighted average of elasticities estimated in the first difference or using a static model (Hendry, 1995[55], see also Table 2).

Table 2 summarizes the estimated elasticities from the error-correction model (ECM) specification as reported in DCSW[24] and compares them with elasticities obtained from the static and first-difference specifications. The semi-elasticities to temperature bins reported in Table 2 suggest that temperature change has an influence on energy demand in 16 out of 24 energy carrier, sector, region combinations. Temperature semi-elasticities indicate that energy carrier demands tend to increase with hot days with a magnitude that varies among energy carriers and sectors. Responses to hot and cold days are asymmetric, and depending on whether an energy carrier is mostly used for heating or cooling, response in one or both directions can be significant. Several semi-elasticities to cold days are negative, suggesting that extreme cold weather could reduce energy demand, especially in production processes in industry and agriculture. The negative estimates can be due to reduced energy consumption for cooling or irrigation during the shoulder seasons of spring and fall (e.g., reduction in electricity use in commercial activities, industry, and agriculture), or to fuel switching (e.g., from petroleum products to electricity in the commercial sector in the tropics). An increase in cold days can temporarily induce production activities to reduce their electricity demand and/or temporarily shift to cheaper sources such as natural gas. Moreover, commercial and industrial consumers may also have back-up energy generation.

In this paper, we use the historical evidence on energy use over a period of about 30 years summarized in Table 2 as an analog of how we might use energy in the future over the next 30 years to generate a set of counterfactual scenarios aimed at exploring climate and socioeconomic uncertainty. Because the elasticities for cold and hot days are estimated individually, future impacts in response to changes in both hot and cold days can be reported individually or combined.

**Baseline projections.** To establish mid-century baseline energy demand in the absence of climate change, we applied income elasticities from DCSW[24] to the increases in countries' GDP per capita from 2010 to 2050 corresponding to each Shared Socioeconomic Pathway (SSP) scenario, generating energy carrier x sector consumption growth factors that were used to scale each sector's per-capita energy demand from its 2010 level. The resulting country-wide average values of per-capita demand for the three energy carriers by four sectors were then combined with gridded maps of future population under the SSPs to yield projected levels of

**Table 3 Key global characteristics of the SSP quantifications for population, GDP, and GDP per capita**

| Indicator | Year | SSP1 | SSP2 | SSP3 | SSP4 | SSP5 |
|---|---|---|---|---|---|---|
| Population | 2010 | 6844 | 6844 | 6844 | 6844 | 6844 |
| (million persons) | 2050 | 8434 | 9137 | 9920 | 9095 | 8532 |
| GDP (trillion | 2010 | 67 | 67 | 67 | 67 | 67 |
| USD$_{2010}$) | 2050 | 285 | 230 | 177 | 219 | 361 |
| GDP per capita | 2010 | 9763 | 9763 | 9763 | 9763 | 9763 |
| (USD$_{2010}$) | 2050 | 33,742 | 25,142 | 17,871 | 24,104 | 42,303 |

energy demand across the globe on a 0.25° grid. Due to the lack of information for spatial distribution of energy demands, the energy demand in each grid cell is simply the national average energy demand per capita multiplied by the population in the grid cell.

The Shared Socioeconomic Pathways (SSPs) that we used in this research have been developed by the climate research community as common basis across mitigation and impact research[28]. The SSPs have diverging narratives that describe how these worlds evolve into high or low challenges to mitigation or adaptation[29]. SSP1 represents a world with low socioeconomic challenges to both adaptation and mitigation. In SSP2, intermediate progresses have been made, and both adaptation and mitigation challenges remain at a medium level, whereas SSP3 represents a future with high challenges on both dimensions. In SSP4 and SSP5, adaptation or mitigation challenges dominate respectively. Several key variables have been projected forward for each of the SSPs. In this paper, we use SSP projections of future spatial population change[30,56] and GDP growth for 183 countries[57]. A global summary of population and GDP of the Shared Socioeconomic Pathways is shown in Table 3 to indicate the wide variation between SSPs. There is some debate on whether each of the five SSPs can actually be combined with all levels of climate change from the Representative Concentration Pathways (RCPs) (for details, see Figure 8 in Riahi et al.[58]). This is especially relevant for the highest-emission scenario, RCP8.5, which can only be reached under SSP5. However, the widest diversion between scenarios, both for the SSPs and the RCPs, takes place in the second half of the 21st century. In this paper, we only focus on results for mid-century, a period for which the potential inconsistency between SSPs and RCPs is much less clear. Moreover, we only use GDP and population from the SSPs and whether the projections for economy and population of 2050 are inconsistent with RCP8.5 levels of warming, is still an open question. Finally, in the structure of our impact analysis, the main difference between the SSPs are different distributions of population over the planet (i.e., different countries have higher/lower population growth) and different sectoral composition of energy demand, due to differences in economic growth. These are both relevant uncertainties to explore in the context of this impact study, and both these issues are not the main uncertainties with respect to (in-)consistency between the SSPs and RCP8.5.

Six Integrated Assessment Modeling (IAM) teams have published energy demand projection for the SSPs[58]. The methods used by these models are fundamentally different from our econometric approach. IAMs provide simplified representations of human and natural systems and integrate the energy systems in the macroeconomic system. Our econometric elasticities do not capture large structural changes in the economy or changes in energy demand patterns over different stages of development. Supplementary Figs. 4 and 5 show a comparison between the range of IAM quantifications of the SSPs and our econometric method (red dots) for each sector, energy carrier combination for the years 2010 (Supplementary Fig. 4) and 2050 (Supplementary Fig. 5).

Because the SSP quantifications of the IAMs are only publicly available for aggregated sectors and regions, we merged our residential and commercial sectors into a single-building sector. Also, there is no SSP information available on the agriculture sector, so we could not make that comparison. For some sector x energy

carrier combinations, we could not estimate statistically significant elasticities, and therefore our method does not project any change from today's level, with future projections of GDP per capita (for instance, for petroleum products in the industry and building sectors). For the largest (and for the fastest-growing) energy carrier $x$ sector combinations, such as electricity in buildings and industry, our baseline demand projections are in line with IAM projections. For some others, such as natural gas in buildings, our method assumes a tighter relation between energy demand and GDP per capita compared with IAMs. For this comparison, it is also important to note that there are some definitional differences for the base year between the IAMs and our data, as can be seen in Supplementary Fig. 4.

**Climate-change impacts.** The last methodological step is calculating the effects of mid-century climate change relative to the baseline. Our main data source is the NASA Earth Exchange Global Daily Downscaled Projections (NEX-GDDP)[34], which tabulate bias-corrected daily maximum and minimum temperatures on a 0.25° grid over the 2006–2100 period for the RCP4.5[26] and RCP8.5[27] scenarios simulated by 21 Earth System models participating in the global Climate Model Intercomparison Project round 5 (CMIP5)[33]. The NASA-NEX-GDDP are both bias correction as well as spatial downscaling of the CMIP5 data to a consistent spatial grid. The bias correction is based on a statistical approach that compares spatially explicit GCM output for a historical period to actual historic data to compute an offset to shift the local GCM results. The spatial downscaling is a combination of linear interpolation, with preservation of the spatial details of the observational data. A detailed description of the NEX-GDDP data set can be found in ref. [34] and https://nex.nasa.gov/nex/static/media/other/NEX-GDDP_Tech_Note_v1_08June2015.pdf.

For each grid cell, we computed mean daily temperatures from the daily max and min temperature provided by the NEX-GDDP data, and then sum the annual days of exposure to average temperatures <12.5 °C and >27.5 °C. In our 2050 no climate-change baseline scenario, each grid cell's vector of temperature exposures was assumed to remain at its 2006–2015 average value. To construct ensemble projections of climate-change impact, we computed the difference between each grid cell's NEX-GDDP projected annual exposure and its baseline exposure over the period 2040–2060 and combined the results with the empirically derived long-run temperature elasticities of energy demand to generate gridded maps of annual changes in energy carrier $x$ sector energy demand, which were then averaged over years to produce the effect of climate circa 2050.

Aggregating the results across energy carriers and sectors yields 21 ESM realizations of future change in total final energy demand for each of two RCP scenarios and five SSP scenarios. The SSP scenarios were developed as discrete storylines with no information as to their relative likelihood of occurrence; accordingly, we treat energy demand under each SSP, as well as its potential to generate a high or low trajectory of radiative forcing, separately. Within each scenario combination, we treat each ESM's gridded realization of temperature as an independent draw from an unknown conditional probability distribution. The realizations of 2050–2010 changes in temperature exposure, and concomitant effects on energy demand, generated by our ESM ensemble can be weighted in any number of ways[59]. The simplicity and transparency of the independence assumption make it a useful jumping-off point for characterizing the risk of climate-change impacts.

## Data availability

The output data generated during our analyses and supporting the findings of this paper are available in the IIASA DARE repository with access code 41: [https://dare.iiasa.ac.at/41/]. This repository also contains R-scripts to regenerate all figures in this paper. The input data used in this analysis are available at the following public locations: NASA-NEX climate data: [https://cds.nccs.nasa.gov/nex-gddp/]. GDP and population for the Shared Socioeconomic Pathways: [https://tntcat.iiasa.ac.at/SspDb]. Spatial population projections for the SSPs: [https://doi.org/10.7927/H4RF5S0P].

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

## Acknowledgements

Climate scenarios used were from the NEX-GDDP data set, prepared by the Climate Analytics Group and NASA Ames Research Center using the NASA Earth Exchange, and distributed by the NASA Center for Climate Simulation (NCCS). BvR was supported through NSF grant No. 1243095, an NCAR-ASP Faculty Fellowship Grant, as well as financial and in-kind support from the Boston University Pardee Center for the Study of the Longer Range Future. The National Center for Atmospheric Research is sponsored by the National Science Foundation under grant No. 929635. ISW was supported by the US National Science Foundation through the Network for Sustainable Climate Risk Management (SCRiM) under NSF cooperative agreement GEO-1240507, and by the U.S. Department of Energy, Office of Science, Biological and Environmental Research Program, and MultiSector Dynamics Program, Grant Nos. DE-SC0005171 and DE-SC0016162. EDC was supported by the ENERGYA project, funded from the European Research Council (ERC) under the European Union's Horizon 2020 research and innovation program under grant agreement No. 756194.

## Author contributions

I.S.W. and B.v.R. conceived the research question and developed the analytical framework. E.D.C. and I.S.W. gathered and processed historical energy and weather data, and generated the empirical estimates. B.v.R. prepared impact projections and follow-on analyses, and manuscript drafts. All authors contributed to refining and editing the paper.

## Additional information

**Competing interests:** The authors declare no competing interests.

