## [Peer Review File · Nature Communications]

Reviewers' comments:

Reviewer #1 (Remarks to the Author):

Amplification of Future Energy Demand Growth due to Climate Change

This manuscript makes use of econometric relationships between national energy demands and temperature in an historical period to estimate changes in energy demand by the middle of this century using warming estimated by 21 earth system models and several socio-economic paths. This reviewer's comments are as follows:

1. The manuscript relies heavily on the econometric results reported in a paper by De Cian and Sue Wing (2018). The manuscript adds more climate and socio-economic scenarios. For this reason, it is important to also comment about the limitations of the paper by Cian & Wing (2018).
2. Cian & Wing use a panel data of multiple countries over the period from 1978 to 2010 to develop econometric relationships between energy demand and temperature. The use of national averaged energy statistics is very problematic for multiple reasons, including the fact that in large countries such as the USA and Russia there are multiple climate zones and the stocks of buildings, appliances, and other energy consuming units are not uniform. For example, in the USA, in states with strong energy efficiency programs like California and New York, energy consumption trends are very different than trends in other states.
3. The use of national temperature time series is also problematic, even if Cian & Wing may have attempted to estimate population weighted national temperature averages. The fact that energy demand as a function of temperature at the local scale is not governed by linear relationships makes any national level econometric analysis for the world highly questionable.
4. Cian & Wing and the manuscript under review estimate elasticities of energy demand for two temperature bins < 12.5 °C and > 27.5 °C. The estimation of only one elasticity parameter per bin is problematic because energy demand responds in a highly non-linear fashion to temperature. Others have used, for example, much more than two bins for temperature at the local scale (e.g., hundreds of grid points in California). The authors implicitly assume that between 12.5 °C and 27.5 °C no energy is consumed. This is not correct. For example, space heating usually starts to increase, at least in the USA, when temperatures are below 16 °C.

5. The manuscript does not include estimation of peak electricity demand, which is an extremely important factor because the electricity system must be designed to satisfy peak electricity demand even when this peak only happens for a few hours/days of the year.

This reviewer believes that more analyses are needed before this manuscript is published as a journal paper. At a minimum, the authors must clearly explain the serious limitations of their method.

Reviewer #2 (Remarks to the Author):

This research focuses on impacts of rising temperature on global energy demand in 2050 covering four end use sectors, agriculture, industry, residential and commercial. It covers three major energy carriers, electricity, natural gas and petroleum-products. Impacts of temperature rise on energy demand is really interesting and it is an important research area to look at. Probably, there is no major global studies that look at the system level available. It is of interest to international communities and can generate much discussion/debate. This is a high-quality work and the paper is well written.

I have couple of major comments:

Why transport was not included? Transport sector energy demand will also be affected by temperature rise due to change in demand for heating and also for air conditioning/cooling. Since this research focuses on final energy (electricity, oil products and natural gas), including transport sector would provide true picture, i.e., changes in final energy demand in various sectors as there are potential for substitutions among various energy carriers (electricity, natural gas and petroleum products) and sectors, transport, agriculture and industry sector. Figure 3 suggests that industry sector benefits from the lower gas demand in residential sector as the Industry sector gas demand increased while the residential sector gas demand decreased under temperature rise scenarios.

This research might partially capture the impact of temperature rise on energy demand as the temperature rise will have impact on productivity. It is unclear whether the impact of heat stress on

productivity included in this study. Heat stress will affect the energy demand due to substitution (labour/machine) in industry and agriculture sector.

Couple of minor comments:

Page 2, line 31-32 says that “rising ambient temperature can decrease cold season heating demand”. But, climate change will have a mixed effect, despite the average temperature rises, cold season still might have some days extremely cold and could affect the heating demand and it also depends on the region.

Page 7, Line 136-137, need some explanation why changes in agriculture sector energy demand will be negative. In page 3, Line 32-33, it says “rising ambient temperatures are expected to increase agricultures demand for irrigation”. But the research finds out that the rising temperature will reduce agriculture sector energy demand.

Reviewer #3 (Remarks to the Author):

This paper presents evidence that climate change impacts will lead to an increase in the demand for electricity to satisfy increasing needs for cooling and heating at global level, albeit with regional differences. This is the first comprehensive study on the topic at the global level and also the first to set the analysis in an uncertainty setting. Highlighting uncertainty in projections is particularly important from a policy point of view, as countries need set up their adaptation and mitigation strategies considering the future uncertainty in climate impacts policy action under uncertainty.

While the paper is well written, some things could be improved and in their economic consequences.

The analysis in the paper is thorough, well explained and convincing. The statistical methods are appropriate and sound. The data sources are also well explained so that potentially the analysis could be reproduced (however not easily as it is not an easy analysis to produce).

The literature on the sectoral economic consequences of climate change has already moved towards uncertainty ranges; especially with the integrated assessment modelling community working on the

Shared Socio-Economic Pathways (SSPs). The paper could definitely highlight the need for uncertainty analysis and help the reader and the wider public understand the main messages of the paper:

- The paper would benefit from highlighting its policy relevance (especially with respect to uncertainty), in the introduction and discussion. The discussion does include some points on the policy relevance of the results but these could be made clearer. It would also be good to state upfront why it is relevant to embark in a huge amount of work on uncertainty in this topic.
- The paper at times uses formulations that are too technical. It would be good to explain the methods in an accessible way and leave technical details for supplementary material.
- It would be good to highlight the role of behavioural uncertainties (not cited in the initial paragraph where the paper gives an overview of sources of uncertainty)
- In the discussion section it would be good to add a discussion of urbanisation. These assumptions are underlying the demographics of the SSPs, but it would still be good to discuss them more explicitly as they influence the results on energy demand and on the possible impacts of climate change on energy demand.
- In the discussion section, it would be good to better discuss the dangers in the increase in energy demand for what regards e.g. energy supply, energy mix for the increased supply (e.g. coal can be increased when weather is warm and there are restrictions to hydro and nuclear) and thus feedbacks to emissions, which may further increase. This can be discussed even if not directly studied in this paper.

Some minor suggestions:

- attention to acronyms
- explain what the SSPs and RCPs are (lines 40-45), e.g. that they have been committed by IPCC to modelling community.
- Line 54: contribution instead of goal
- Line 71: "By combining the income elasticities with SSPs": I see it says "See Methods", but it is still necessary to explain in an easier way what the paper is doing. The phrase is really unclear.
- Lines 74-75: "We superimpose...", could be explained in a more accessible way in the text + refer to annex.
- Line 77: spell out "ESMs" for first time.
- Line 91: more recent numbers instead of 2010?
- Line 104: 96-250 EJ-> can you make this number easier to relate to? E.g. is it p.c.? what is it equivalent to?

- Line 164: “a minority of the population experiences modest...” -> where does this happen?

- The paper often refers to IAMs and the IAM community; it would be good to make explicit the fact that these results are also very relevant for computable general equilibrium models used to study the economy-wide consequences of climate impacts, as these can take into consideration also the indirect effects on different sectoral and international economic flows.

Reviewers' comments:

Reviewer #1 (Remarks to the Author):

Response: We thank the reviewer for laying out five important points of discussion that we have fully taken into account in the revised manuscript. Building on the reviewer's suggestions, we have further expanded the discussion on limitations, added an explicit comparison between bottom-up and top-down econometric estimates to the SI, and expanded on the potential use of the proposed methodology. Below we provide a point-to-point response and we refer to where in the manuscript the text has been changed accordingly.

Amplification of Future Energy Demand Growth due to Climate Change

This manuscript makes use of econometric relationships between national energy demands and temperature in an historical period to estimate changes in energy demand by the middle of this century using warming estimated by 21 earth system models and several socio-economic paths. This reviewer's comments are as follows:

1. The manuscript relies heavily on the econometric results reported in a paper by De Cian and Sue Wing (2018). The manuscript adds more climate and socio-economic scenarios. For this reason, it is important to also comment about the limitations of the paper by Cian & Wing (2018).

Response: In this paper, we build on the peer-reviewed and published methods from De Cian & Sue Wing (2018), which belong to a well-established stream of literature. However, we use the concerns raised by this reviewer to include an elaborate discussion (in the Discussion Section) to the extent that the shortcomings of these methods are relevant to the application in this manuscript.

We take this opportunity to mention that our approach follows a well-established literature in energy and environmental economics. The validity of this approach has been established by a sizable peer-reviewed literature, including economics (e.g. Dell et al. 2014 JEL, Deschenes and Greenstone 2012 AER) and interdisciplinary (Burke et al. 2018 Nature) top journals. The abovementioned examples apply econometric or statistical methods to aggregate national statistics and to estimate historically based, reduced-form response functions. The main goal of this approach is explore the scale and the geographic pattern of projected climate change impacts, while being explicit about the sources of uncertainty that are being taken into account.

2. Cian & Wing use a panel data of multiple countries over the period from 1978 to 2010 to develop econometric relationships between energy demand and temperature. The use of national averaged energy statistics is very problematic for multiple reasons, including the fact that in large countries

such as the USA and Russia there are multiple climate zones and the stocks of buildings, appliances, and other energy consuming units are not uniform. For example, in the USA, in states with strong energy efficiency programs like California and New York, energy consumption trends are very different than trends in other states.

Response: There is a delicate balance or tension in our analyses between the ability to be globally comprehensive and to include all the relevant details. On that, we agree: the use of national fuel demand as an explanatory variable is subject to numerous limitations, on which we have added a detailed discussion in the Discussion section. We do agree that one would prefer to use locally estimated elasticities before applying the results of this study for local planning and have added a discussion on this to the paper. Indeed “different horses should be used for different courses”, and that an aggregate, long-term analysis at global level, like the one proposed in this paper, does not fit all purposes. We agree that such approach cannot be used to address questions related to how design the electricity system in order to deal with peak loads in a specific location (see our response to the next comment). Yet, our approach is a valuable and valid method to inform people, governments, investors, multi-nationals operating across different countries regarding the scale of potential projected climate changes. It is a valuable and valid method to alert policy makers focusing on mitigation regarding the potential pressure of climate-induced energy demand on emissions, the energy system, and therefore the decarbonization effort. Our results on how climate change will redistribute energy demand geographically, seasonally across sectors and income groups can inform long-term, high-level (e.g. UNFCCC) decisions, can be used as input in further analyses that aim at examining the implications for the economy, sustainable development, and the environment. Shorter-term decisions on local planning and management of energy systems would need to look at results from country or location-specific studies.

Even so, apart from EU member states and the USA, national-level annual total final consumption of fuels is simply the best - in fact the only - energy demand data that are available. The data needs of studies that look into specific countries or states, are not compatible with data availability at the global level. A growing number of data products are attempting to increase the spatial resolution of macro statistics with global coverage, such as GDP per capita, agricultural productivity, but a similar effort is still missing with respect to energy data. Moreover, the validity of using downscaled data would need to be established, as results would very much depend on the downscaling algorithm.

3. The use of national temperature time series is also problematic, even if Cian & Wing may have attempted to estimate population weighted national temperature averages. The fact that energy demand as a function of temperature at the local scale is not governed by linear relationships makes any national level econometric analysis for the world highly questionable.

Response: Estimation of temperature semi-elasticities from the covariation between national average per capita fuel demand and population-weighted temperature exposures possibly suffers from aggregation bias. Here again, the issue is the pervasive lack of data. Attempting to capture local heterogeneity in the climate response by running time-series regressions for each country would run into the problem of limited numbers of observations, which would still preclude identification of the heterogeneous marginal effects of an additional day of exposure in multiple temperature intervals. De Cian and Sue Wing (2019)—hereafter DCSW—initially employ a vector of temperature bins in an attempt to capture precisely the effect described above (see their Fig. 2A). However, due to multicollinearity the coefficients on multiple adjacent bins were not precisely estimated. To address this problem, adjacent bins were successively merged to arrive at a specification for which the effects of exposure to heat were identified. This is the crux of the tradeoff between fidelity and coverage.

4. Cian & Wing and the manuscript under review estimate elasticities of energy demand for two temperature bins < 12.5 °C and > 27.5 °C. The estimation of only one elasticity parameter per bin is problematic because energy demand responds in a highly non-linear fashion to temperature. Others have used, for example, much more than two bins for temperature at the local scale (e.g., hundreds of grid points in California). The authors implicitly assume that between 12.5 °C and 27.5 °C no energy is consumed. This is not correct. For example, space heating usually starts to increase, at least in the USA, when temperatures are below 16 °C.

Response: The reviewer’s comment belies a misinterpretation of DCSW’s results that turns out to be crucial to reconciling their estimates with those of more fine-grained climate econometrics studies. Upon exposure to temperatures 12.5 °C $< T < 27.5$ °C a country’s demand is not zero—it doesn’t respond significantly to temperature and remains at its conditional mean per capita level determined by idiosyncratic factors and income. This is a critical point, on which the reconciliation of DCSW’s coarse-resolution “top-down” estimates with fine-resolution “bottom-up” estimates by, e.g., Auffhammer et al (2017)—hereafter ABH—depends.

The limitation of DCSW’s specification is that it constrains each additional hot day or cold day to have the same effect (respectively), irrespective of the average temperature of that day was close to the cutoffs or far into the tails of the temperature distribution. It is the potential nonlinearity of the response in the tails of the temperature distribution that is likely the source of the reviewer’s concern.

The new section S1.3 of the Supplementary Information provides such a reconciliation, based upon a simplification of ABH Fig. 1, shown below as Fig R1. We here briefly discuss this reconciliation and its implications, but refer to SI section S1.3 to a more detailed description.

We draw particular attention to the fact that even electric power demand responses that are characterized using many bins of temperature, and estimated on large numbers of observations at fine spatial- and temporal scales, can still be well approximated within the estimated confidence intervals by linear schedules for increasing high temperatures and decreasing low temperatures, outside of an intermediate moderate temperature zone. This point is emphasized by ABH. Using the control points from the figure (Fig R1), we recast ABH’s specification as a piecewise-linear degree-day spline (see eq. (S2), parameterized in Table S6). Using this approximation, we demonstrate that three factors have the potential to drive a wedge between ABH’s local and DCSW’s geographically averaged estimates: local weather-insensitive energy consumption (i.e., the actual electricity demand that occurs in ABH’s omitted 15 °C $< T < 18$ °C bin), the sub-national distribution of population and the distribution of average daily temperatures above and below DCSW’s hot and cold cutoffs.

We test of how different the joint response of PJM and ERCOT is from DCSW’s estimates is, using data for ABH’s 2006-2014 study period on county population and air temperature. As summarized in Table S7, the two sets of temperature semi-elasticities are of the same general magnitude, but, compared to ABH, DCSW’s weighted average response to cold days is smaller (not significantly different from zero), while their response to hot days is larger.

Fig R1: Daily electricity temperature response functions from Auffhammer et al., 2017

We then examined the implications of these differences for projections of impact. We developed a method to compare the impacts of ABH's total demand in levels against DCSW's sectoral demand in logarithms based on a consistent partitioning of total annual demand into a weather-responsive component and a weather-insensitive component based on similar sub-national conditional electric loads.

Calculations of the demand impact for the 2090-2099 future period relative to the 2006-2015 current period for our simplified version of ABH's model closely align with those reported in ABH Table 1. Impacts based on DCSW's estimates are overestimated in ERCOT, underestimated in PJM, and exhibit more variance, especially for counties exposed to higher temperatures. We conclude that despite the limitations discussed in the text and quantified in the SI, DCSW's semi-elasticities form a credible basis for projecting the impacts of climate change on energy demand.

5. The manuscript does not include estimation of peak electricity demand, which is an extremely important factor because the electricity system must be designed to satisfy peak electricity demand even when this peak only happens for a few hours/days of the year.

Response: Recent studies (Auffhammer et al, 2017; Wenz et al, 2017) highlight the importance of climate warming impacts on peak electricity demand. While the infrastructure implications of fine-temporal-scale responses of electricity demand to temperature are certainly interesting and important in their own right, and increases in total electricity use associated with cooling are central to the increases in demand we project, specifically estimating how peak as opposed to total demand might increase is outside the scope of this study. On line 295-300 of the text we are careful to emphasize that while climate warming exerts a positive amplification effect of on total energy demand over and above that associated with the growth of population and income, irreducible uncertainty in the magnitude of this influence will pose a challenge to energy infrastructure planning over the coming decades. Given the aforementioned limitations of the empirical estimates that underlie our projections, we are hesitant to extend our analysis beyond projecting increases in total demand. Moreover, lack of hourly electric load data for countries outside Europe and the USA precludes the development of internationally comparable empirical estimates of peak demand response. And while one might be tempted to obtain answers by transferring Auffhammer et al's and Wenz et al's estimates to other geographic domains, the points that the reviewer him/her-self makes regarding the specificity of local demand responses argue strongly against the validity of such an approach. Therefore, attempting to project fine temporal scale increases in demand at the level of individual fuels and/or sectors over multiple decades would require too specific details at the global scale is fraught with too much uncertainty.

This reviewer believes that more analyses are needed before this manuscript is published as a journal paper. At a minimum, the authors must clearly explain the serious limitations of their method.

Response: We thank the reviewer for pushing us to undertake additional analysis that has strengthened the paper, especially in ways that may assist others to reconcile bottom-up impact estimates with those developed from our top-down approach.

We have expanded the section in the Discussion to highlight the consequences of the underlying econometric methods on the results of this paper and added an elaborate section to the SI that reconciles the econometric estimates from different time/geographic scales. We have stated more clearly upfront in the paper the pros and cons of using this method, added an elaborate discussion on the caveats of this global analysis to the Discussion Section. We clarify that our approach can be used to show how climate change impacts will redistribute energy consumption in multiple ways, geographically, seasonally, across income groups, in a manner that depends on the climate and socio-economic uncertainties. We clarify that this is the first step to look at issues related to technological and behavioral uncertainty that can be better explored in a subsequent step using IAMs or CGE models.

Reviewer #2 (Remarks to the Author):

This research focuses on impacts of rising temperature on global energy demand in 2050 covering four end use sectors, agriculture, industry, residential and commercial. It covers three major energy carriers, electricity, natural gas and petroleum-products. Impacts of temperature rise on energy demand is really interesting and it is an important research area to look at. Probably, there is no

major global studies that look at the system level available. It is of interest to international communities and can generate much discussion/debate. This is a high-quality work and the paper is well written.

Response: Thank you very much for your kind words

I have a couple of major comments:

Why was transport not included? Transport sector energy demand will also be affected by temperature rise due to change in demand for heating and also for air conditioning/cooling. Since this research focuses on final energy (electricity, oil products and natural gas), including transport sector would provide a true picture, i.e., changes in final energy demand in various sectors as there are potential for substitutions among various energy carriers (electricity, natural gas and petroleum products) and sectors, transport, agriculture and industry sector. Figure 3 suggests that the industry sector benefits from the lower gas demand in the residential sector as the industry sector gas demand increased while the residential sector gas demand decreased under temperature rise scenarios.

Response: Transport was included in the original paper of De Cian and Sue Wing (2018) because that was requested by a reviewer of that paper but we realized that the identified elasticities could be capturing secondary trends that could not be fully attributed to the relation between hot/cold days and energy use. Although there is good reason to expect a relation between temperature and energy use in the transport sector, temperature-related energy use is marginal for transportation purposes and we decided to focus on the final sectors where temperature more directly affects the climate-sensitive energy use to guarantee the thermal comfort of people. The dynamics of the transport sector are more complex. Temperature traditionally does not enter the model of the demand for transportation, and the dynamics would be sub-sector specific (e.g. freight transport would have different dynamics compared to personal mobility choices, in the residential sector demand for mobility services would also interact with leisure-labor choices). Moreover, the way energy use in the industrial, commercial, agricultural sector is defined matters. For example, energy use in agriculture includes not only energy use for irrigation, but it also includes energy consumed by agriculture, hunting and forestry whether for traction (excluding agricultural highway use), power or heating (agricultural and domestic).

This research might partially capture the impact of temperature rise on energy demand as the temperature rise will have an impact on productivity. It is unclear whether the impact of heat stress on productivity included in this study. Heat stress will affect the energy demand due to substitution (labour/machine) in the industry and agriculture sector.

Response: The reviewer could be correct that the underlying elasticities from De Cian and Sue Wing (2018) have picked up an impact of temperature on productivity. This potential impact could be positive or negative in sign, depending on the climatic zone and type of activity. However, given that the scope of this study is limited to the implications of temperature changes on energy use, we have not further explored this interesting side-way. We focus on adaptation via changing energy use to ensure thermal comfort without an explicit link to the benefits in terms of reduced mortality, morbidity, or productivity. Yet, mitigating heat stress is clearly one of the primary reasons for using energy to maintain thermal comfort. Also, with this framework we are not able to say anything regarding adaptation effectiveness, as despite the use of energy to cope with heat or cold, residual impacts might persist, and further affect labour productivity. The only variable that might be capturing this in a very indirect way would be GDP per capita. With our current datasets, it would not be trivial to quantitatively identify such a relationship at global scale, though we are aware of ongoing studies exploring this at regional scale (e.g. for the EU).

Couple of minor comments:

Page 2, line 31-32 says that “rising ambient temperature can decrease cold season heating demand”. But, climate change will have a mixed effect, despite the average temperature rises, cold season still might have some days extremely cold and could affect the heating demand and it also depends on the region.

Response: we have added a nuancing ‘could’ to this sentence, but feel that including this caveat would be too much detail for the opening sentence of the paper

Page 7, Line 136-137, need some explanation why changes in agriculture sector energy demand will be negative. In page 3, Line 32-33, it says “rising ambient temperatures are expected to increase agricultures demand for irrigation”. But the research finds out that the rising temperature will reduce agriculture sector energy demand.

Response: Thanks for this good catch. We have added an explanatory sentence to the manuscript. The main reason is that Wilbanks and Fernanez only mention electricity, which is mainly used for irrigation (which increases in both cases), whereas our analysis also includes a decrease in natural gas and petroleum products.

Reviewer #3 (Remarks to the Author):

This paper presents evidence that climate change impacts will lead to an increase in the demand for electricity to satisfy increasing needs for cooling and heating at global level, albeit with regional differences. This is the first comprehensive study on the topic at the global level and also the first to set the analysis in an uncertainty setting. Highlighting uncertainty in projections is particularly important from a policy point of view, as countries need set up their adaptation and mitigation strategies considering the future uncertainty in climate impacts policy action under uncertainty.

Response: Thank you very much for your kind words

While the paper is well written, some things could be improved and in their economic consequences. The analysis in the paper is thorough, well explained and convincing. The statistical methods are appropriate and sound. The data sources are also well explained so that potentially the analysis could be reproduced (however not easily as it is not an easy analysis to produce).

Response: Thank you very much for your kind words

The literature on the sectoral economic consequences of climate change has already moved towards uncertainty ranges; especially with the integrated assessment modelling community working on the Shared Socio-Economic Pathways (SSPs). The paper could definitely highlight the need for uncertainty analysis and help the reader and the wider public understand the main messages of the paper:

- The paper would benefit from highlighting its policy relevance (especially with respect to uncertainty), in the introduction and discussion. The discussion does include some points on the policy relevance of the results but these could be made clearer. It would also be good to state upfront why it is relevant to embark in a huge amount of work on uncertainty in this topic.

Response: Thank you for these supportive comments. We have added a framing of the relevance of uncertainty to the introduction and discussion sections.

- The paper at times uses formulations that are too technical. It would be good to explain the methods in an accessible way and leave technical details for supplementary material.

Response: We have reformulated several key-sentences to be less technical. We would opt to keep the detailed description of the methodology in the Methods section, though, mostly in the light of the comments by Reviewer 1.

- It would be good to highlight the role of behavioural uncertainties (not cited in the initial paragraph where the paper gives an overview of sources of uncertainty)

Response: Thanks, we indeed overlooked mentioning that source of uncertainties and have added it to the introduction. We have also expanded the discussion on the importance of technological and structural change, two sources of uncertainty that we do not explore in this paper because, as we write in the introduction, we feel this approach is not the right way to do it. We argue that uncertainties related to changes in structural, technological and market characteristics, as well as price-induced behavioral changes can be better analyzed using IAM or CGE frameworks, starting from the ex-ante shocks provided by this paper.

- In the discussion section it would be good to add a discussion of urbanisation. These assumptions are underlying the demographics of the SSPs, but it would still be good to discuss them more explicitly as they influence the results on energy demand and on the possible impacts of climate change on energy demand.

Response: We have added the following discussion on urbanization:

Urbanization is implicit in our methodology, as future temperature fields are population-weighted, and the spatial distribution of population varies across SSPs along with country-level population trends, which do play a large role in the underlying SSP projections. Across the SSPs, urbanization levels vary widely, with low urbanization in SSP3 and high urbanization in SSP1 and SSP5. Urbanization drives the location of population in the SSP spatial population projections, where they are allocated from the national level to individual grid cells. In our projections, urbanization determines to which temperature changes people are exposed, and temperature changes in urban grid-cells are more dominant higher urbanization scenarios.

- In the discussion section, it would be good to better discuss the dangers in the increase in energy demand for what regards e.g. energy supply, energy mix for the increased supply (e.g. coal can be increased when weather is warm and there are restrictions to hydro and nuclear) and thus feedbacks to emissions, which may further increase. This can be discussed even if not directly studied in this paper.

Response: the original manuscript discussed this topic briefly, but we have now expanded that discussion as follows:

As early as mid-century there can be benefits to mitigating climate change (t-tests indicate a statistically significant 14-20% difference in the means of the distributions) but the associated costs depend critically on the uncertain greenhouse gas intensity of electricity generation that satisfies the anticipated large increases in future demand. Such increased electricity consumption does not translate one-to-one in carbon emissions, as, depending on the merit order, higher or lower carbon intense electricity production technologies might be dominant.

Some minor suggestions:

- attention to acronyms

Response: done

- explain what the SSPs and RCPs are (lines 40-45), e.g. that they have been committed by IPCC to modelling community.

Response: done

- Line 54: contribution instead of goal

Response: done

- Line 71: "By combining the income elasticities with SSPs": I see it says "See Methods", but it is still necessary to explain in an easier way what the paper is doing. The phrase is really unclear.

Response: done

- Lines 74-75: "We superimpose...", could be explained in a more accessible way in the text + refer to annex.

Response: done. This description of the method had become too condensed and we have expanded it.

- Line 77: spell out "ESMs" for first time.

Response: done

- Line 91: more recent numbers instead of 2010?

Response: we mention 2010 because it is the "baseyear" of our analysis. The temperature and GDP per capita changes that drive the changes in energy demand are computed for the difference between 2010 and 2050.

- Line 104: 96-250 EJ-> can you make this number easier to relate to? E.g. is it p.c.? what is it equivalent to?

Response: As the global average per capita climate-exposed energy consumption it not a very intuitive number either (nor is the change in that indicator between present-day and mid-century, given changes in population size), we have rephrased this sentence to include the present-day level of global climate-exposed energy consumption:

Globally, 2050 baseline energy demand is two to three times larger than today (Table 1), or an increase from 137 EJ of climate-exposed final energy use in 2010 to 234-388 EJ by mid-century (see Table S3).

- Line 164: "a minority of the population experiences modest..." -> where does this happen?

Response: we have added a sentence stating this is mostly in Russia, Western Europe, Canada and Chile/Argentina

- The paper often refers to IAMs and the IAM community; it would be good to make explicit the fact that these results are also very relevant for computable general equilibrium models used to study the economy-wide consequences of climate impacts, as these can take into consideration also the indirect effects on different sectoral and international economic flows.

Response: we have added reference to the CGE modeling community as well, for whom these projections may be even more relevant.

Reviewers' comments:

Reviewer #1 (Remarks to the Author):

Thank you for seriously considering my comments. This reviewer recommends publishing the paper.

Reviewer #2 (Remarks to the Author):

Authors responded to the two major comments explaining why they did not include transport and heat stress in the response to feedback. But, no changes were made to the manuscript. Since this paper discuss overall energy demand and trade-offs among sectors and fuels, I think the transport sector is important as it can affect the overall final energy mix. The manuscript did not even have a word "transport". Similar to people occupied in buildings, people occupied in vehicles also need a level of comfort and therefore the vehicle energy consumption will be affected due to climate change. At least, the authors should mention the importance of transport sector in the manuscript and clearly explain why it was not included in the study.

Reviewer #3 (Remarks to the Author):

Dear editor,

I find that the authors have written a very detailed report on the comments I had submitted and addressed all main comments in the revised manuscript.

When they did not make changes to address a comment, they have clearly explained why they chose not to do so.

Therefore, I find that the revised manuscript is ready for publication.

Kind regards.

Reviewer #2 (Remarks to the Author)

Authors responded to the two major comments explaining why they did not include transport and heat stress in the response to feedback. But, no changes were made to the manuscript.

Since this paper discuss overall energy demand and trade-offs among sectors and fuels, I think the transport sector is important as it can affect the overall final energy mix. The manuscript did not even have a word “transport”. Similar to people occupied in buildings, people occupied in vehicles also need a level of comfort and therefore the vehicle energy consumption will be affected due to climate change. At least, the authors should mention the importance of transport sector in the manuscript and clearly explain why it was not included in the study.

Response: We have added a paragraph mentioning the importance of transport and explaining why it was not included in this study. As explained on page 12, line 263-269, our projection methodology cannot discern the fraction of energy that is sensitive to variations in weather and therefore it assumes that over the long run all energy consumption is responsive to temperature. Since temperature-related energy use is marginal for transportation purposes, we decided to focus on the other sectors. Here is the text we have no included in the discussion section, page 13, line 299.

Future climate change is likely to affect energy use in the transport sector as well, but, as suggested in DCSW, the underlying drivers remain opaque. On the one hand, people occupying vehicles modify cooling and heating while driving to maintain thermal comfort. On the other hand, people faced with extreme cold or hot weather conditions might simply make different mobility choices (e.g. drive more, or do not drive at all). Weather traditionally does not enter the model of the demand for transportation and temperature-related energy use is marginal for transportation purposes. Since our projection methodology cannot discern the fraction of energy that is sensitive to variations in weather, we focus on the final sectors where temperature more directly affects the climate-sensitive energy use. The dynamics of the transport sector are more complex and sub-sector-specific (e.g. freight transport would have different dynamics compared to personal mobility choices, in the residential sector demand for mobility services would also interact with leisure-labor choices), calling for dedicated studies first identifying the underlying mechanisms at play, and subsequently evaluating them empirically.

REVIEWERS' COMMENTS:

Reviewer #2 (Remarks to the Author):

Authors have added a paragraph explaining why transport was not included in the study and therefore I recommend it for publication.

REVIEWERS' COMMENTS:

Reviewer #2 (Remarks to the Author):

Authors have added a paragraph explaining why transport was not included in the study and therefore I recommend it for publication.

Response: thank you